# DeepSTABp: A Deep Learning Approach for the Prediction of Thermal Protein Stability

**DOI:** 10.3390/ijms24087444

**Published:** 2023-04-18

**Authors:** Felix Jung, Kevin Frey, David Zimmer, Timo Mühlhaus

**Affiliations:** Computational Systems Biology, RPTU University of Kaiserslautern, 67663 Kaiserslautern, Germany

**Keywords:** imbalanced dataset, protein language model, protein stability prediction, protein structure analysis, protein melting point, thermal proteome profiling

## Abstract

Proteins are essential macromolecules that carry out a plethora of biological functions. The thermal stability of proteins is an important property that affects their function and determines their suitability for various applications. However, current experimental approaches, primarily thermal proteome profiling, are expensive, labor-intensive, and have limited proteome and species coverage. To close the gap between available experimental data and sequence information, a novel protein thermal stability predictor called DeepSTABp has been developed. DeepSTABp uses a transformer-based protein language model for sequence embedding and state-of-the-art feature extraction in combination with other deep learning techniques for end-to-end protein melting temperature prediction. DeepSTABp can predict the thermal stability of a wide range of proteins, making it a powerful and efficient tool for large-scale prediction. The model captures the structural and biological properties that impact protein stability, and it allows for the identification of the structural features that contribute to protein stability. DeepSTABp is available to the public via a user-friendly web interface, making it accessible to researchers in various fields.

## 1. Introduction

Protein stability refers to the ability of a protein to maintain its structural and functional integrity under various environmental conditions [1,2,3]. While different environmental factors can affect protein stability, thermal stability is an important property of proteins, as many biological processes occur at specific temperatures. Proteins that are less thermally stable are more prone to aggregate at physiological temperatures leading to loss of activity, dysfunction or even the formation of toxic protein aggregates [1,4]. Thermal stability can be influenced by a number of factors including amino acid composition, protein–protein interaction, post-translational modifications, and the presence of ligands or other molecules [4,5]. Understanding the thermal stability of a protein can be important for various applications, such as biotechnology and food science, where proteins are often exposed to changes in temperature during cultivation, processing, and storage [4,5,6]. The thermal stability of a protein can be measured by its denaturation or melting temperature (*T*_m_), which is the temperature at which 50% of the protein loses its native structure and activity or alternative defined as the area under the melting curve. In the past, measuring the stability of proteins required extensive work and resulted in limited data. However, with the advancement of mass spectrometry-based thermal proteome profiling (TPP), it is now possible to simultaneously monitor the thermal stability of thousands of expressed proteins in a variety of settings such as in vitro, in situ, or in vivo. TPP is based on the principle that when proteins are exposed to heat, they denature and become insoluble. The proteins that remain in the soluble fraction at different temperatures are quantified by protein mass spectrometry and combined across the temperatures to generate individual melting profiles [7,8].

Making use of a mass spectrometry-based proteomic approach, an atlas of the thermal stability could be compiled that comprises 48,000 proteins across different species ranging from archaea to humans and covering melting temperatures of 30–90 °C [9]. This meltome atlas has proven to be an invaluable resource for biological research and drug discovery. However, due to technical limitations, cost, and the labor-intensive nature of the experimental approach, the proteome and species amenable to thermal profiling are limited [7,8,9]. As a result, predicting protein thermal stability has become a modern solution when experimental data is incomplete or when assessing thermal stability is not easy. There two major classes of protein stability predictors which either directly predict the stability of a protein or forecast how sequence changes affect a given stability [10,11,12,13]. In order to be able to augment TPP experimental data, we are focusing on the direct prediction of the protein *T*_m_. For this purpose, several prediction methods have been developed that utilize a variety of features describing proteins based on their amino acid composition, physicochemical and surface properties, statistical and sequence potentials, and other protein characteristics.

In the past, building stability prediction models faced the challenge of limited data availability [13,14,15,16]. However, the current challenge has shifted towards larger training datasets that are biased by dominant species, resulting in the underrepresentation of crucial properties and protein characteristics. Additionally, reductionistic or predefined feature selection negatively impacts model performance because stability is a complex property with multiple contributing features. Further, available algorithms for thermal stability prediction based on cell-wide analysis of protein stability TPP experiment differ regarding the definition of the learning problem. DeepTP [17] and BertThermo [18] approaches construct a classification problem to distinguish between thermostable and thermolabile proteins, but do not intend to predict *T*_m_ values. Although classification-based predictors have demonstrated outstanding performance, they simplify the prediction task by reducing the number of output classes to a discrete set, whereas regression-based analysis captures the continuous nature of *T*_m_ values. However, defining the task as a regression problem also presents additional challenges, such as non-uniformity of recorded datasets and gaps in the modelled domain.

To address these challenges, we present a novel machine learning architecture, DeepSTABp, for the reliable regression-based prediction of cellular thermal protein stability. DeepSTABp is based on a transformer-based protein language model for amino acid sequence embedding and state-of-the-art feature extraction. It combines different popular deep learning techniques into a standalone end-to-end protein *T*_m_ predictor. Our model incorporates experimental conditions used in the TPP experiment, the protein amino acid sequence, and the organism growth temperature. By using an appropriate sampling strategy, we trained the model for large-scale prediction and achieved significantly better results than current approaches such as ProTstab2 [13,16], which has been the leading method for *T*_m_ value prediction so far. Further, we can show that the resulting augmented TPP dataset captures structural and biological properties and allows us to dissect the structural features that impact protein stability.

## 2. Results

### 2.1. Data Driven Design of the DeepSTABp Model

Assessment of the thermal stability of proteins was long hindered by limited available experimental methodologies. In recent years, the rise of mass spectrometry-based thermal proteome profiling (TPP) has allowed for the creation of proteome-wide snapshots of protein melting points (*T*_m_s) [7]. Extensive efforts have resulted in the detection of tens of thousands of protein melting characteristics from a wide variety of organisms across a broad range of growth temperatures [9]. However, our analysis of existing datasets revealed that many of the proteins of said organisms remain unaccounted for (Figure 1). Evaluation of these results not only motivated but also guided the design of a novel protein melting point predictor which we termed DeepSTABp (Figure 2).

The following three design decisions were directly informed by the analysis of the dataset: (i) In similar fashion to previously published *T*_m_ predictors, we decided to incorporate protein primary structure information into the model [13,16]. However, we did not rely on classical methods of protein feature extraction but instead decided to utilize the transformer-based protein language model ProtTrans as one of DeepSTABp’s central building blocks. This allowed our model to be trained on protein sequence embeddings provided by ProtTrans which were optimized through the analysis of millions of protein sequences, overcoming the limitations of our TPP dataset, which only consists of 35,112 unique protein sequences (see Section 4). (ii) Protein stability is not only determined by protein sequence but also other organism-specific factors related to the cellular environment, such as pH, presence of ions, chaperones or PTMs, to name just a few [1]. In order to account for such factors and backed by the observed correlation between organism-specific growth temperature and average protein melting point in TPP datasets, the model was extended by an MLP block (Figure 1B and Figure 2). (iii) Additionally, our assessment of available datasets led to the introduction of a third neuronal network block which allows one to account for the experimentally observed differences in protein complex behavior in TPP datasets acquired from cell lysates and whole-cell measurements [9] (Figure 2).

### 2.2. Evaluation and Comparison of Prediction Performance

The analysis of available TPP data made apparent that protein *T*_m_s do not follow a uniform distribution on an organism level. Rather, protein *T*_m_s distributions possess non-uniform shapes, ranging from only slightly skewed Gaussian-like distribution centered at organism-specific *T*_m_s, to multimodal distributions [9]. Since it has been previously shown that the training of neuronal networks is prone to be negatively affected by non-uniform distributed training datasets, we decided to measure the effect of different sampling strategies termed “naïve”, “uniform sampling” and “extended uniform sampling” with respect to model performance (Table 1, [19,20,21,22]).

TO assess model performance, we computed 5 common metrics for the evaluation of regression models: The coefficient of determination (R^2^), the Sample Pearson correlation coefficient (PCC), the mean average error (MAE), the mean squared error (MSE), and the root mean squared error (RMSE). When evaluating prediction results for the training dataset, all metrics indicate that model performance is indeed impacted by the choice of the training dataset, with the best result achieved consistently when using the EUS upon dataset assembly.

Finally, we evaluated DeepSTABp’s performance with respect to generalizability and in comparisons of *T*_m_ predictions of the current state-of the-art predictor ProTstab2, which emerged out of a thorough comparative evaluation of different machine learning frameworks and has been shown to provide best-in-class performance [13]. The resulting data indicates that DeepSTABp generalizes well when applied on the previously unseen datasets and does not suffer from substantial overfitting. Moreover, our model outperforms ProTstab2 substantially in all evaluation metrics during training and when applied to a previously unseen test dataset (Table 1). Visualizing the distributions of *T*_m_ differences between predicted and experimentally determined melting temperatures, it becomes apparent that both models show a bell-like shape, however DeepSTABp’s predictions show a lower overall error as indicated by a strongly reduced interquartile range, spanning only an interval of ~4.9 °C compared to ProTstab2s with ~8.0 °C (Figure 3B).

### 2.3. Capture and Representation of Biological Features

Although regression evaluation metrics are a valuable means of measuring the performance of a model’s predictions, it can be challenging to determine the extent of biological information conveyed by the achieved level of prediction quality. In order to determine whether patterns observed in the analysis of TPP data can be replicated with the attained level of prediction accuracy, we calculated the variation in *T*_m_ for protein complexes annotated in the EMBL complex portal measured via TPP from *Homo sapiens* (571 protein complexes) and *E. coli* (82 protein complexes) versus a randomly sampled sets of proteins of respective organisms [23,24] (Figure 4).

Indeed, our results show that the coefficients of variation estimated from measured and predicted *T*_m_s possess similar distributions along proteins reportedly present in protein complexes. This observation is also in good agreement with previously conducted TPP studies where it was demonstrated that proteins present in the same protein complex show similar thermal proximity coaggregation profiles [23]. This comparison indicates that the precision of DeepSTABp predictions allows them to resolve biological meaningful properties.

### 2.4. Future Applications of DeepSTABp

The assessment of available proteomic data revealed that a substantial portion of proteomes are still not covered by TPP. Thus, the use of artificial intelligence to bridge this gap and increase the available information by predictions rather than experimental measurements remains an attractive endeavor. Here, we examined whether the *T*_m_ predictions of our model for *Homo sapiens* allow for the identification of differences in the distributions of aggregated protein structure elements that change relative to predicted temperature stability (Figure 5).

The analysis revealed that protein-wise aggregates of protein secondary structure elements “AlphaHelix”, “BetaSheet”, “PiHelix”, “Turn”, “NoStructure” and “Accessible Protein Surface” show different patterns when related to predicted *T*_m_s. While proteins with low (red) to high (blue) content of alpha helixes are found across the whole *T*_m_ range of the human proteome, other properties show more inhomogeneous shapes. In the case of “PiHelix” and “Accessible Protein Surface”, we see an antiparallel pattern. While proteins with highest amounts of pi-helices are more frequently found at higher melting points, the opposite is true for the latter. Both observations fit well with previous reports, as pi-helices and protein packing density were indeed linked to an improvement in thermal stability [25,26,27,28].

In conclusion, the results indicate that the combination of in silico methods, namely the *T*_m_ predictions by DeepSTABp and protein structure prediction-based inference of protein features allows for the identification of patterns which were previously linked to protein stability by experimental studies.

### 2.5. Web-Interface of DeepSTABp

The utilization of DeepSTABp to predict protein *T*_m_s for research purposes does not necessitate any installation procedures or substantial computational resources. Rather, users may acquire protein-specific predictions through a user-friendly web interface accessible at https://csb-deepstabp.bio.rptu.de (accessed on 4 April 2023) (Figure 6). The process consists of three primary steps: (i) the provisioning of a protein sequence or protein database in the FASTA format, (ii) selection of a growth temperature and (iii) setting the ‘lysate’ or ‘cell’ flag. Predicted Tms are supplied to the user in a human and machine-readable text file. The web service allows to process protein sequences of arbitrary length and is suitable for genome-wide studies. The source code of the webservice is openly accessible (https://github.com/CSBiology/deepStabP (accessed on 12 April 2023)) and provided as docker images, which renders the tool readily deployable within on-premise solutions. Alternatively, DeepSTABp can also be run on a desktop pc. The average execution time for a protein sequence of 1000 amino acids is on average 17 s on an AMD Ryzen^Tm^ 9 6900HS or half a second using an NVIDIA GeForce RTX 3070 Ti.

## 3. Discussion

Protein stability plays a crucial role in a variety of biological processes and biotechnological applications. The thermodynamic stability of a protein determines its folding, assembly and function, and therefore, changes in stability can have significant effects on protein properties and are frequently observed to be disease related. Hence, studying protein stability is an essential aspect of protein biochemistry and structural biology. Due to the sparsity of experimental data, computational tools are necessary to predict protein stability given in the form of melting temperatures.

The deep learning-based predictor DeepSTABp presented in this study employs a protein language model to embed amino acid sequences and advanced feature-extraction techniques. It integrates various deep learning methods into a self-contained protein *T*_m_ predictor. In contrast to other predictors like SCooP, DeepTP, and BertThermo essentially classifying thermostable and non-thermostable proteins, Prostab2 and DeepSTABp both employ a regression model to predict continuous *T*_m_ values [13,15,16,17,18]. Through proper sampling techniques, we trained the model for extensive prediction tasks and achieved superior results compared to the current state-of-the-art method, ProTstab2, by reducing the mean average prediction error by around 35 percent (Table 1, MAE). DeepSTABp relies on a pretrained protein language model that allows for unbiased, nonreductive feature extraction from protein amino acid sequence. It has been shown that the size of the training dataset correlates with predictive power of the language model effectively learning structure-relevant features that would be not possible to extract from the TPP dataset alone according to size limitations [29]. Further, our model incorporates the experimental conditions used in the TPP experiment and the organism growth temperature, leading to improvement of the prediction quality. The beneficial effect of incorporating growth temperature into our model suggests that there are organism-specific confounder variables like proteins or metabolites acting as chaperones or compatible solutes, respectively, influencing thermostability [30,31,32,33].

Our prediction has enriched the TPP dataset, enabling us to unravel the structural and biological determinants of protein stability. Of particular interest is the observation that proteins exhibiting low surface accessibility tend to be found at higher temperatures, which fit with previous reports of protein packing density positively influencing thermal stability [25,26,27].

The high-quality predictions of DeepSTABp have enabled us to investigate the biologically relevant formation of protein complexes, replicating the known evolutionary phenomenon whereby proteins within complexes exhibit similar Tm values. This highlights the potential for future models to incorporate information about protein complexes in order to enhance prediction accuracy. Going forward, we plan to incorporate additional experimental data to refine our model. Although our protein Tm prediction performance has been excellent overall, we acknowledge the model’s limitation in accurately predicting the effect of single point mutations on global protein stability. This is a common issue faced in protein property prediction that needs to be addressed and analyzed in the future [34]. However, the modular design of DeepSTABp allows for the exchange of the transformer block and the addition of new building blocks. Therefore, further research in the area of global protein property prediction can be integrated effortlessly to overcome this limitation.

In summary, our DeepSTABp algorithm is characterized by its high speed and reliability, making it well-suited for the analysis of proteins from diverse organisms and large-scale protein datasets. Our testing encompassed proteomes from a broad range of species, spanning from archaea to humans, and encompassing melting temperatures available in the meltome atlas. To support the in silico exploration of protein thermostability, we have made the DeepSTABp model available to other researchers through a user-friendly web interface (https://csb-deepstabp.bio.rptu.de (accessed on 4 April 2023)).

## 4. Materials and Methods

### 4.1. T_m_ Dataset Assembly and Extraction of Protein Melting Points

Datasets used for model training and evaluation in this study were derived from high-throughput mass spectrometry-based thermo-proteome profiling (TPP) assays. To achieve an extensive and homogenous collection of experimentally determined protein melting temperatures (*T*_m_s), we reanalyzed obtained melting curves and determined individual protein melting points by fitting the following non-linear model:(1)$$F_{soluable}(T)=\frac{1-a}{1+e^{-m(\frac{1}{T}-\frac{1}{T_{mid}})}}+a$$
with *a* being the asymptote, *m* being the slope, and *T*_mid_ denoting the mid-temperature of the fitted curve [35]. The *T*_m_ was obtained by finding the temperature where the fitted function reaches a value of 0.5. In order to retrieve only reliable, high quality *T*_m_s, only model estimates with an R^2^ score > 0.9 and temperature variance across biological replicates < 2 degrees were retained in the final data set.

The vast majority (21,885 ids) of protein *T*_m_s was derived from the meltome atlas [9], which not only provides a collection of TPP datasets derived from a multitude of different organisms but also consists of TPP studies involving the heat treatment of cells and proteins lysates. Since the originally available dataset lacked the preparation of *S. cerevisiae* cells and *Homo sapiens* lysates, we extended the atlas with data from recently published studies [23,36]. The final dataset consisted of 35,112 proteins originating from *Escherichia coli*, *Saccharomyces cerevisiae*, *Oleispira antarctica*, *Arabidopsis thaliana*, *Drosophila melanogaster*, *Caenorhabditis elegans*, *Mus musculus*, *Homo sapiens*, *Thermus thermophilius* and *Picrophilus torridus*. Protein sequences were acquired from Uniprot [37]. Prior to model training, the assembled dataset was randomly split into training (90%) and validation (10%) data sets. In order to allow for a fair comparison to the previously published predictor Prostab2, we additionally created a test dataset which resembled the test dataset used in their respective study and excluded said proteins from training procedures.

### 4.2. Dataset Augmentation

We created 3 distinct training datasets by tuning the “naïve” dataset through 2 different sampling strategies, which we will refer to as “uniform sampling” (US) and “extended uniform sampling” (EUS). The US sampling strategy is carried out independently for each organism in the training data set. It uses a simulated continuous uniform distribution limited to the minimum and maximum *T*_m_ of the respective organism. Each *T*_m_ in the original dataset is then replaced by a random draw out of the simulated distribution. The EUS strategy samples a total of 10,000 training items for each organism. All datasets are made publicly available in accordance to the FAIR principles and provided within the Annotated Research Context (ARC) available at: https://git.nfdi4plants.org/f_jung/deepstabp (accessed on 12 April 2023).

### 4.3. Amino Acid Sequence Embedding Using Protein Language Models

To obtain a feature space spanned by numerical variables, protein sequences are often preprocessed by feature extraction routines that generate numerical embeddings of amino acid sequences. For the development of our algorithm, we chose to deploy the transformer-based protein language model ProtT5-XL-UniRef50 released as part of the ProtTrans project (available at: https://github.com/agemagician/ProtTrans (last accessed on 5 March 2023)). This model has been shown to be an efficient and competitive feature extractor, which allowed for the construction of state-of-the-art predictors achieving best-in-class performance in tasks like protein localization or membrane prediction [29,38]. Regarding Tm prediction, we could show that its deployment led to an improved prediction performance in comparison to feature extraction as utilized in the ProTstab2 model (Appendix A)

### 4.4. DeepSTABp

In this study, we treated *T*_m_ prediction as a regression problem with the goal of deriving a model that relates a feature space spanned by the variables, protein sequence, growth optimum and experimental design to the dependent variable, the melting temperature of a protein. To achieve this, the DeepSTABp algorithm (Figure 2) combines different popular deep learning architectures into a single stand-alone end-to-end protein *T*_m_ predictor. The final model was made available in the ONNX format and released alongside the source code for model training and evaluation in accordance with the FAIR principles and provided as an annotated research context (ARC) available at: https://git.nfdi4plants.org/f_jung/deepstabp (accessed on 12 April 2023).

The model architecture is split into four blocks. The first block was included to incorporate whether TPP experiments were carried out following the heating of intact cells or cell lysate, respectively, as this factor has been previously shown to affect measured *T*_m_ values of certain proteins [9,23]. It consists of two separate, fully connected layers with 20 and 10 neurons, respectively. The second block is concerned with sequence embedding (see: amino acid sequence embedding using protein language models). It uses the pretrained ProtT5-XL-UniRef50 model, a transformer-based network architecture which creates a 1024 entry long feature vector for each amino acid in the input sequence. As a final step, this block contains a mean pooling layer which allows the final model to process sequences of arbitrary length [9].

These networks are accompanied by a third block, an independent neural network that also consists of a sequence of 2 fully connected layers with 20 and 10 neurons, respectively. This block uses frequently reported growth temperatures of the organisms of interest as an input which allows the model to learn how factors that are independent of protein sequence influence a proteins *T*_m_ (see dataset assembly and extraction of protein melting points, Figure 2 [39,40,41,42,43,44,45,45,46,47,48,49,50]). Finally, the outputs of the first third blocks are concatenated and used as the input of the fourth block, which consists of a sequence of five dense, fully connected layers with 4098, 512, 256, 128, and 1 neurons, respectively. Neurons of the fully connected layers were modeled as scaled exponential linear units (SELUs) in order to minimize the risk of vanishing gradients during model training [51]. This was followed by layer normalization to ensure that all features were on the same scale, which prevents exploding gradients and speeds up the training [52]. To minimize the chance of overfitting, all MLP blocks were trained using the “dropout” technique, which randomly (*p* = 0.2) mutes the influence of single neurons with respect to the computation of gradients and thus parameter tuning during the training phase [53]. The optimization of the network weights of the final model was carried out using the Adam optimizer with a flexible learning rate which was adjusted after 10 epochs without significant changes in model performance. The choice of optimal network and training parameters was guided by hyperparameter tuning using a tree-structured Parzen estimator (TPE) sampling [54] algorithm. As final parameters we chose an epoch size of 100, a batch size of 25, and a starting learning rate of 0.01.

### 4.5. Model Evaluation Metrices

To validate the performances of models during training and testing and to allow for a fair comparison to alternative approaches, different commonly used evaluation metrics were computed. Each metric measures the discrepancy between vectors of N experimentally determined *T*_m_s (**y**) and predicted *T*_m_s (**ŷ**). In summary, we computed five different metrics:

The mean average error (MAE):(2)$$MAE(y,\hat{y})=\frac{1}{N}\sum_{i=1}^{N}\left|y_{i}-{\hat{y}}_{i}\right|$$

The mean squared error (MSE):(3)$$MSE(y,\hat{y})=\frac{1}{N}\sum_{i=1}^{N}{\left(y_{i}-{\hat{y}}_{i}\right)}^{2}$$

The root mean squared error (RMSE):(4)$$RMSE\left(y,\hat{y}\right)=\sqrt{MSE(y,\hat{y})}$$

Sample Pearson correlation coefficient (PCC):(5)$$PCC\left(y,\hat{y}\right)=\frac{n\sum_{i=1}^{N}y_{i}{\hat{y}}_{i}-\sum_{i=1}^{N}y_{i}\sum_{i=1}^{N}{\hat{y}}_{i}}{\sqrt{n\sum_{i=1}^{N}y_{i}^{2}-{\left(\sum_{i=1}^{N}y_{i}\right)}^{2}}\sqrt{n\sum_{i=1}^{N}{\hat{y}}_{i}^{2}-{\left(\sum_{i=1}^{N}{\hat{y}}_{i}\right)}^{2}}}$$

And the coefficient of determination (R^2^):(6)$$R^{2}\left(y,\hat{y}\right)=1-\frac{\sum_{i}^{N}{\left(y_{i}-{\hat{y}}_{i}\right)}^{2}}{\sum_{i}^{N}{\left(y_{i}-\overset{-}{y}\right)}^{2}}$$

### 4.6. Variation of Measured and Predicted T_m_s of Protein Complexes

Since it was previously reported that proteins present in complexes possess similar thermal proximity coaggregation profiles, we aimed to analyze whether this is observation is reflected by *T*_m_s determined experimentally and through prediction by DeepSTABp [23]. To annotate which proteins were present in protein complexes, we retrieved a collection of curated protein complexes for *E. coli* and *Homo sapiens* from the EMBL-Complex Portal [24]. For each of the selected organisms, we iterated our final *T*_m_ dataset and grouped proteins according to the collected complex annotation. To measure the *T*_m_ similarity within each group, we computed the coefficient of variation (cv) as follows:(7)$$cv\left(y\right)=\frac{\sqrt{\frac{1}{N}\sum_{i=1}^{N}{\left(y_{i}-\overset{-}{y}\right)}^{2}}}{\overset{-}{y}}$$
with y being the collection of grouped protein *T*_m_s, *N* being the number of proteins present in the group and $\overset{-}{y}$ being the average within-group *T*_m_. As a control, we took a total of 10 random samples per protein in each protein complex group and computed the cv thereof.

### 4.7. Relating Predicted T_m_s to Secondary Structure Elements

Protein structure predictions for *Homo sapiens* were retrieved from the AlphaFold Protein Structure Database [55,56]. The resulting PDB file for each protein containing the coordinates of each AA in a three-dimensional space were then used as the input for the DSSP program to assign secondary structure elements of each protein [57,58]. Following the assignment of protein secondary structure properties to each residue by DSSP, we aggregated the properties “AlphaHelix”, “BetaSheet”, “PiHelix”, “Turn” and “NoStructure” protein-wise by expressing their abundance relative to the peptide length. As the importance of protein packing was previously related to protein stability, we calculated a feature termed “Accessible Protein Surface”, by summing up the “Accessible Surface” determined by DSSP for each amino acid residue. In order to relate said features to predicted protein *T*_m_s for *Homo sapiens*, we chose to first sort each protein into a *T*_m_ bin with the bin size determined using Silverman’s rule for Gaussian kernel density estimation applied to the whole dataset [59]. Subsequently, we computed the quartiles of each protein property using the complete dataset. For each *T*_m_ bin, we counted the number of proteins in the bin belonging to a quartile, yielding 4 *T*_m_ distributions per protein feature.

### 4.8. Software

Model training and evaluation was implemented in the Python programming language (v.3.10.8) using the PyTorch (v.1.13.0), PyTorch Lightning (v.1.7.7), pandas (v.1.5.1), NumPy (v.1.23.4), and Transformers (v.4.24.0) library [60,61,62,63,64]. The Optuna library (v.3.0.5) was utilized for conducting hyperparameter tuning [65]. Statistical calculation regarding secondary structure elements was conducted using the open-source library FSharpstats (v.0.4.7) and BioFSharp (v.2.0.0) [59,66]. Data visualization was achieved using the Plotly and Plotly.NET library (v.3.0.0) [67,68]. Feature extraction was conducted with protr [69].

## Figures and Tables

**Figure 1 ijms-24-07444-f001:**
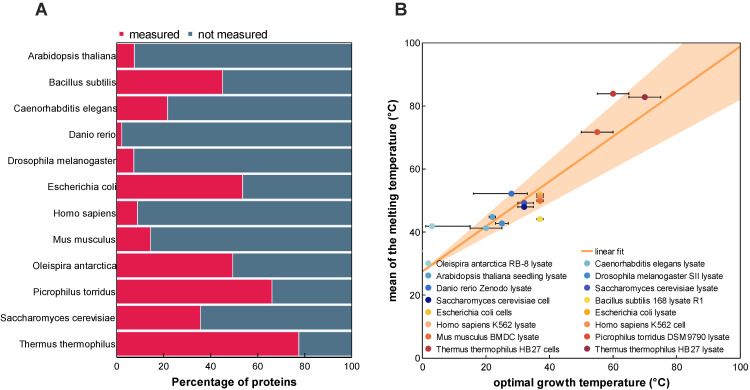
Melting temperature coverage (**A**) and relationship with the optimal growth temperature (**B**). (**A**) Comparison between the number of proteins measured in the meltome atlas and the number of proteins attributed to an organism according to the reference proteome. (**B**) Relationship between the melting temperature (*T*_m_) and the optimal growth temperature of the twelve organisms covered in the analyzed TPP datasets. Ranges of the optimal growth temperature for each organism are indicated with error bars. Marked points denote the temperatures that were used for the optimal growth temperature feature during model training and evaluation.

**Figure 2 ijms-24-07444-f002:**
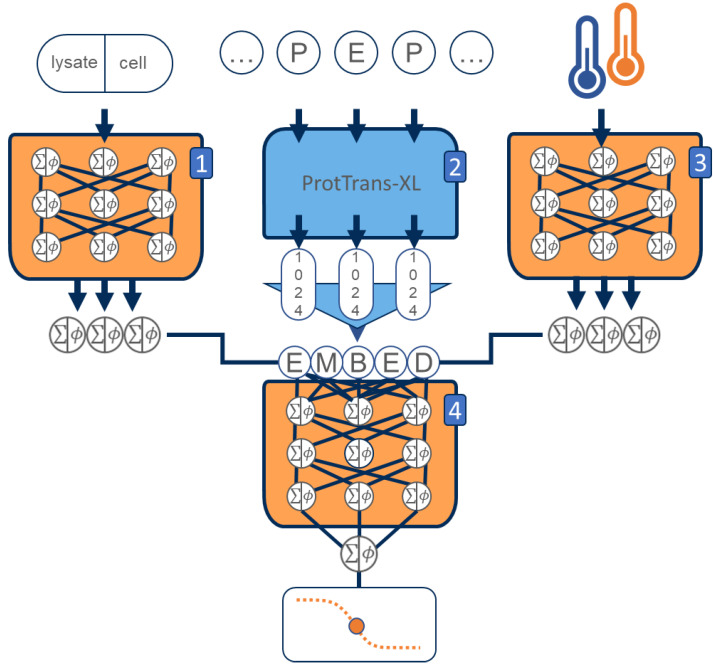
Schematic overview of the deep learning architecture DeepSTABp trained to predict protein melting temperatures. The proposed model is based on four different artificial network blocks. The first three blocks are responsible of creating an embedding of the protein query based on the input features, (1) type of experimental condition used in the thermal proteome profiling experiment, (2) the protein amino acid sequence and (3) the organism growth temperature. While Blocks 1 and 3 use small multilayer perceptrons (MLP), Block 3 consists of the pretrained transformer-based model ProtTrans-XL, followed by a mean pooling layer. The output vectors of the first blocks are concatenated and then used as inputs for a (4) final MLP block, whose output is the predicted protein *T*_m_.

**Figure 3 ijms-24-07444-f003:**
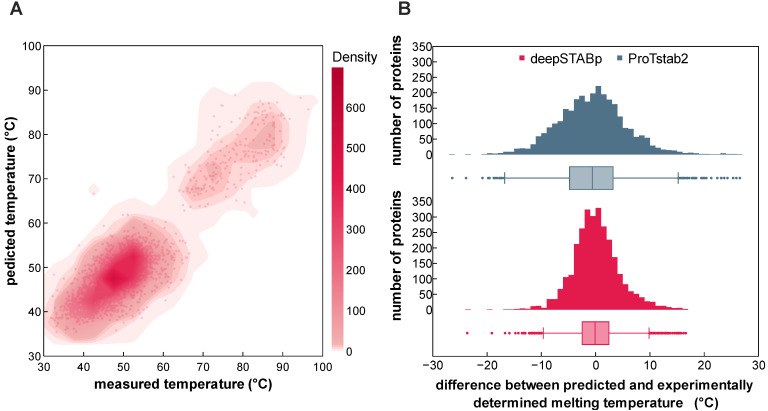
Prediction performance on the test dataset. Visualized are the predicted melting temperature of DeepSTABp against experimentally determined melting temperatures (**A**) and a direct comparison of the prediction performance of DeepSTABp and the current state-of-the-art predictor ProTstab2 shown as difference between predicted and experimentally determined melting temperatures (**B**).

**Figure 4 ijms-24-07444-f004:**
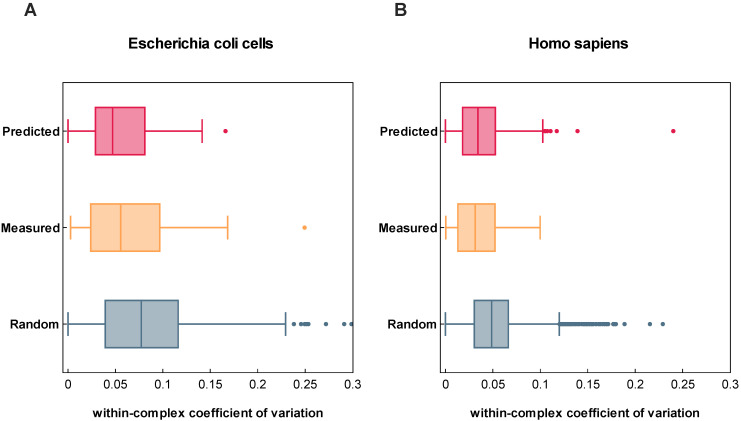
Variation of measured and predicted melting temperatures (*T*_m_s) of protein complexes and randomly grouped proteins of *E. coli* (**A**) and *Homo sapiens* (**B**). The calculated coefficient of variation based on analysis of measured and predicted *T*_m_s show a solid agreement for both organisms. As previously described in the analysis of thermal proteome profiling data, both reflect the tendency that proteins present in protein complexes show a lower variation in thermostability when compared to randomly grouped proteins.

**Figure 5 ijms-24-07444-f005:**
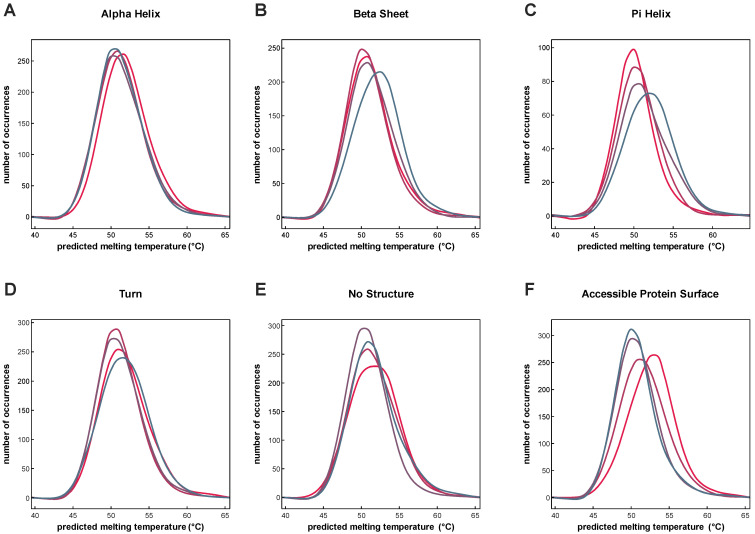
Analysis of dependencies between protein melting temperature and the secondary structure elements AlphaHelix (**A**), BetaSheet (**B**), PiHelix (**C**), Turn (**D**), NoStructure (**E**), and the Accessible Protein Surface (**F**). After binning each protein according to its *T*_m_, each protein was classified according to its value in the respective property into quartiles. Quartile one (grey), quartile two (purple), quartile three (dark pink), and quartile four (red). Observed distributions of protein-wise aggregates show distinct shapes with small (AlphaHelix) to large differences (e.g., Accessible Protein Surface) in the quartile-wise distributions.

**Figure 6 ijms-24-07444-f006:**
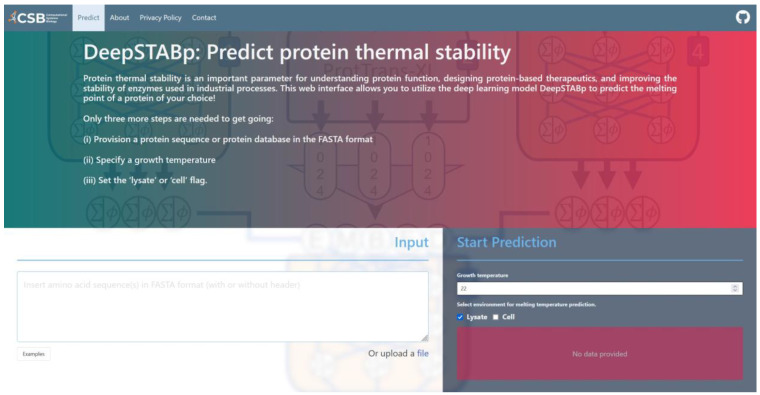
Screen capture of the web-interface of DeepSTABp. The screenshot shows how the DeepSTABp model can be accessed via a user-friendly web interface. After provisioning of input parameters, researchers are provided with predicted protein melting temperatures.

**Table 1 ijms-24-07444-t001:** Assessment of model performance during model training and testing. During training, we monitored model performance with respect to different sampling strategies used during dataset assembly. With “extended uniform sampling” yielding the best results, we built DeepSTABp and evaluated its performance in direct comparison to ProTstab2.

Metrics	Training (DeepSTABp)			Testing	
	Naïve	Uniform Sampling	Extended Uniform Sampling	DeepSTABp	ProTstab2
R^2^	0.86	0.89	0.93	0.80	0.57
PCC	0.93	0.96	0.97	0.90	0.76
MAE (°C)	3.20	2.43	1.81	3.22	4.95
MSE (°C)	17.84	9.70	5.54	18.46	41.59
RMSE (°C)	4.22	3.11	2.35	4.30	6.45

## Data Availability

All datasets are made publicly available at: https://git.nfdi4plants.org/f_jung/deepstabp (accessed on 12 April 2023).

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
