# Peer review of "DeepSTABp: A Deep Learning Approach for the Prediction of Thermal Protein Stability"

_ijms, 2023, doi:10.3390/ijms24087444_

Round 1

Reviewer 1 Report

Several approaches to the thermal stability of proteins, according to their Tm parameter of denaturation, are currently being developed. T. Mühlhaus’s deepSTABp method is interesting.

 The authors clearly advocate for a better and more precise approach than the competitor proTstab, reporting comparisons in similar conditions, with a clear description of what they do.

I have no objection about its publication as it is, except of course for some typos the Editor would be able to correct easily, mainly in the "references" section ( a blank in ref.10, caps in refs 44 and 48..)

Also would be good to put the open access web URL at the end of the summary.

Author Response

Response to the reviewers’ comments

Reviewer: 3

In this manuscript, the authors propose the new predictor DeepSTABp to predict the thermal stability of proteins. I have no significant comments. The work is well done. The manuscript is written correctly, the methods are well-chosen, and the results are clearly presented. This study is interesting I am not an expert in every aspect of this research, but based on my knowledge, I think the work deserves attention. The work may be published in its current form.

We thank the reviewer for the positive evaluation of our work.

Reviewer 2 Report

In this study, Jung et al. proposed a thermal protein stability predictor based on a deep learning method called DeepSTABp. This method can predict the thermal stability of a wide range of proteins and is a powerful and effective tool for large-scale prediction. Experimental results show that the model yields better performance than other methods. I have a few questions and suggestions about this manuscript.

1.     Instead of relying on classical protein feature extraction methods, the authors decided to utilize the pre-trained protein language model ProtTrans as a feature learning module for DeepSTABp. How can it be demonstrated that the features learned using the pre-trained transformer-based model are more helpful in terms of model performance than those extracted by classical protein feature extraction methods, perhaps supplemented by relevant ablation experimental results should be provided.

2.     How did the authors combine different features as input to the last MLP? Are there any other feature fusion methods considered?

3.     In addition to relevant ablation experiments, useful features learned by DeepSTABp can be demonstrated by visualizing the features learned by different modules, for example, using t-SNE visualization.

4.     What are the advantages and limitations of the prediction problem of thermal protein stability, defined as a regression problem, compared to defining it as a classification problem? Please explain in the introduction section.

5.     How did the authors optimize the parameters of the model, using a grid search or some other way? The information would probably be better provided in a table.

6.     The website is very nice and simple to layout and use, but perhaps it would be more helpful if you could provide some examples for users to use.

7.     Please reorganise Table 1, it is not suitable for publication.

Reviewer 3 Report

In this manuscript, the authors propose the new predictor DeepSTABp to predict the thermal stability of proteins. I have no significant comments. The work is well done. The manuscript is written correctly, the methods are well-chosen, and the results are clearly presented. This study is interesting I am not an expert in every aspect of this research, but based on my knowledge, I think the work deserves attention. The work may be published in its current form.

Author Response

(The authors gave the same response as above.)
